# Discrete Element Modelling of a Bulk Cohesive Material Discharging from a Conveyor Belt onto an Impact Plate

**Otto C. Scheffler** †[ID] **and Corné J. Coetzee** *,†[ID]

Department of Mechanical & Mechantronic Engineering, Stellenbosch University, Stellenbosch 7600, South Africa; ocscheffler@gmail.com
* Correspondence: ccoetzee@sun.ac.za; Tel.: +27-(0)21-808-4239
† These authors contributed equally to this work.

**Abstract:** The discrete element method (DEM) has become the numerical method of choice for analysing and predicting the behaviour of granular materials in bulk handling systems. Wet-and-sticky materials (WSM) are especially problematic, resulting in build-up and blockages. Furthermore, due to the large number of particles in industrial-scale applications, it is essential to decrease the number of particles in the model by increasing their size (upscaling or coarse graining). In this study, the accuracy with which upscaled DEM particles can model the discharge of a cohesive material from a belt conveyor onto an inclined impact plate was investigated. Experimentally, three sand grades (particle size distributions, PSDs) were used, each in a dry (non-cohesive) state and with three levels of moisture-induced cohesion. The effects of the modelled PSDs on the material flow, build-up on the plate, the peak impact force and the residual weight were investigated. Although a linear cohesion contact model was mostly used, the results were also compared to that of the Johnson–Kendall–Roberts (JKR) and simplified JKR (SJKR) models. It was found that the general profile of the pile (build-up) could be accurately modelled, but using a more accurate (but still upscaled) PSD improved the results. The impact force and the residual weight on the plate could be accurately modelled (error < 15%) if the particle size was not excessively scaled. The maximum acceptable scaling factor was found to be a geometric factor of the bulk measure of interest, and not a factor of the physical particle size. Furthermore, with an increase in cohesion, the bulk measures such as the thickness of the discharge stream and the height of the material build-up increased, which meant that the maximum acceptable scale factor also increased. The results are valuable for future accurate and efficient modelling of large industrial scale applications of WSMs.

**Keywords:** DEM; validation; cohesion; conveyor transfer; build-up; particle scale

## 1. Introduction

In the agricultural, food processing, pharmaceutical, and mining industries, the pivotal role of equipment like screw conveyors, belt conveyors, bucket elevators, and bucket wheels in the transportation of bulk granular materials cannot be overstated. Belt conveyors are often used due to their versatility and efficiency in transporting large quantities of materials across extended distances. However, the intricate layout of industrial facilities, coupled with changes in terrain and direction, necessitates the incorporation of transfer points [1].

The primary function of a transfer point is to accept material from the feeding conveyor and facilitate its transition onto the receiving conveyor. In designing transfer points, several factors must be taken into account. Examples include the central and symmetric loading of the material onto the receiving conveyor, minimising the angle and velocity at which the material impacts the chute and the receiving belt, the prevention of spillage and blockages, and any spatial constraints such as the maximum height difference between the two conveyors [2].

Analytical and semi-empirical models [3–6] for analysing conveyor transfers are limited in their capabilities [7–12]. DEM, however, provides the analyst with more information

and can easily model complex three-dimensional material flow. A demonstration of this by Nordell in 1994 [13] was one of the earliest, while the implementation of DEM as a predictive design tool by Hustrulid in 1996 and 1998 [14,15] was also among the first.

Several studies demonstrated the use of DEM in modelling a transfer chute and showcasing its capabilities, but did not present any validation of the results [1,2,16–22]. Other studies compared the DEM results to those of analytical models, but not to experimental results [12,23–27], while others performed industrial-scale case studies which were validated qualitatively but not quantitatively [25,28–33].

Lastly, there are only a relatively small number of detailed studies where the results from DEM were compared to laboratory-scaled measurements. These experiments provide excellent opportunities for model validation since material flow rates can be accurately set and monitored, load cells can be positioned to measure impact forces and high-speed imagery can be used to visualise the flow [34], all without interrupting the commercial activities of an industrial plant. Chen et al. [35] analysed a stacker chute and compared the DEM results to scaled laboratory tests; however, only qualitative comparisons were made. Hastie and Wypych [12,23,24] modelled the discharge of non-cohesive materials from a head pulley and through a hood-and-spoon transfer chute. The flow trajectory was compared to measurements and analytical models. Grima [33,36] analysed the flow of non-cohesive materials onto an impact plate, including the effects of modelled particle shape and size. Ilic [37] investigated the loading profile of non-cohesive and cohesive materials on a belt conveyor. The results were compared to laboratory measurements and site tests. The behaviour of the material in the belt transition zone and the discharge trajectory were also analysed. Carr et al. [38–42] modelled wet-and-sticky material (WSM) discharging from a conveyor belt onto an impact plate as part of calibrating the input parameters. Using a liquid-bridge contact model for particle–particle contacts and a simplified Johnson–Kendall–Roberts (SJKR) model for particle–wall contacts, the build-up of material on the plate was analysed in terms of mass (weight) and shape (profile). Ilic and Katterfeld [43] used contact models and calibration techniques similar to Carr et al., and compared the build-up of wet gypsum on an inclined impact plate. Qualitatively, a good correspondence between DEM and experimental results could be achieved in terms of the amount of build-up and the profile of the heap that formed. Rossow and Coetzee [34] analysed the flow of a non-cohesive material through a transfer point with chevron-patterned belts. This included an impact plate, hood, rock box, a chute and the use of particle image velocimetry (PIV). To reduce computational costs, different particle shapes (spheres and multi-spheres) and upscaled PSDs were analysed. Once calibrated, the particle shape had no significant effect; however, there was a limit to which the particles could be scaled. In terms of impact plate forces, the particles could be increased in size by a factor ranging from 7 to 9. However, the material build-up in a rock box and a chute could only be accurately modelled if the scaling factor was less than 1.6.

None of the existing studies modelled the discharge of a cohesive material from a conveyor belt onto an impact plate, in combination with systematic particle upscaling and comparing the results of material build-up and impact forces to laboratory measurements and observations. In this study, a cohesive material with calibrated parameter values, published by the same authors [44,45], was analysed and the effects of particle upscaling investigated. A guideline for the maximum particle scale factor that still ensures accurate results while minimising computational effort is provided. This is valuable for future modelling of large industrial-scale applications.

## 2. Materials and Methods

The cohesive material, laboratory conveyor system, and the DEM model are discussed.

### 2.1. Cohesive Materials

Three grades of Silica sand were used, as shown in Figure 1. The coarse grade had particles ranging in size from approximately 3 mm to 5 mm (weighted mean average

diameter $\bar{d}_p = 4.75$ mm), the medium grade from 1 mm to 3 mm ($\bar{d}_p = 1.92$ mm), and the fine grade from 0.002 mm to 1 mm ($\bar{d}_p = 0.29$ mm).

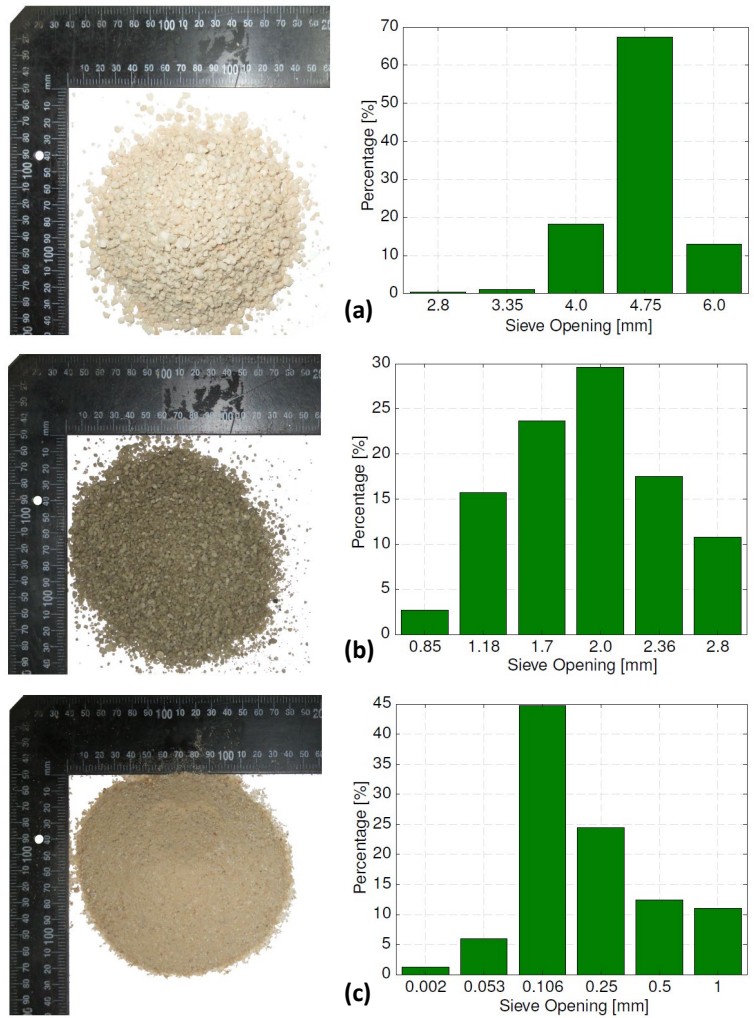

**Figure 1.** Sand grades showing (**a**) the coarse grade, (**b**) the medium grade and (**c**) the fine grade. Reproduced with permission from [45]; published by Elsevier, 2023.

There are various techniques available to measure moisture content, and here, the level of saturation $S$ was used. In the dry state, the saturation is $S = 0\%$, and when $S = 100\%$, all pores are fully filled with water. Three levels of cohesion were induced by adding moisture (water) to the sand such that $S = 5\%$, $S = 10\%$ and $S = 15\%$, respectively. This is the same sand as was used in two related studies; for more details, see [44,45].

*2.2. Conveyor System*

The recirculating conveyor system is shown in Figure 2a. The two units were standard off-the-shelf items from the construction industry. Both conveyors were identical and measured 7.2 m between the head and tail pulley centres. The conveyor belts consisted of a smooth rubber surface with a width of 350 mm. The pulley diameters were equal and measured 220 mm, and the idler troughing angle was 30 deg. Both conveyors were inclined at 15 deg, and chutes were used to guide the material from one conveyor to the other to ensure continuous circulating flow of the material.

The impact plate shown in Figure 2b was placed at the end of one of the conveyors. The lower edge of the impact plate was positioned 500 mm directly below the centre of the pulley (Figure 3a). The plate measured 600 mm × 600 mm and was equipped with load cells to measure the impact load. An HBM S2M 200 N load cell was used to measure the

shear load and a Revere 363 500 N load cell to measure the normal load, both sampled at 200 Hz. A slightly noisy force signal was observed, which could be attributed to the elasticity of the impact plate assembly and the vibrations caused by the running conveyors. Consequently, a moving-average filter (5% span) was employed. For more design details, see Rossow and Coetzee [34].

The plate was inclined at 32° to coincide with the lowest angle of repose measured for any of the sand grades used [44,45]. To eliminate any unknown material–plate interface properties, the plate was fitted with 30 equally spaced fins (17.5 mm high). Prior to any testing, a layer of the sand was poured between the fins and carefully levelled (Figure 2c). With the fins keeping the layer of sand in place, subsequent material flow interacted with the material layer and not with the plate itself. The same procedure was used in setting up the DEM model (Figure 3b).

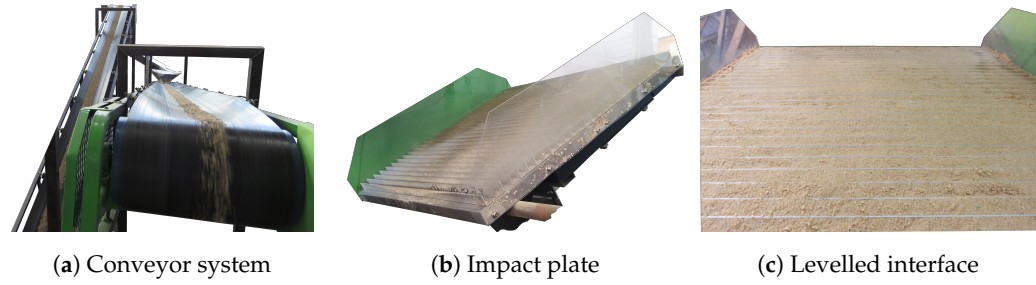

(**a**) Conveyor system      (**b**) Impact plate      (**c**) Levelled interface

**Figure 2.** The experimental conveyor system.

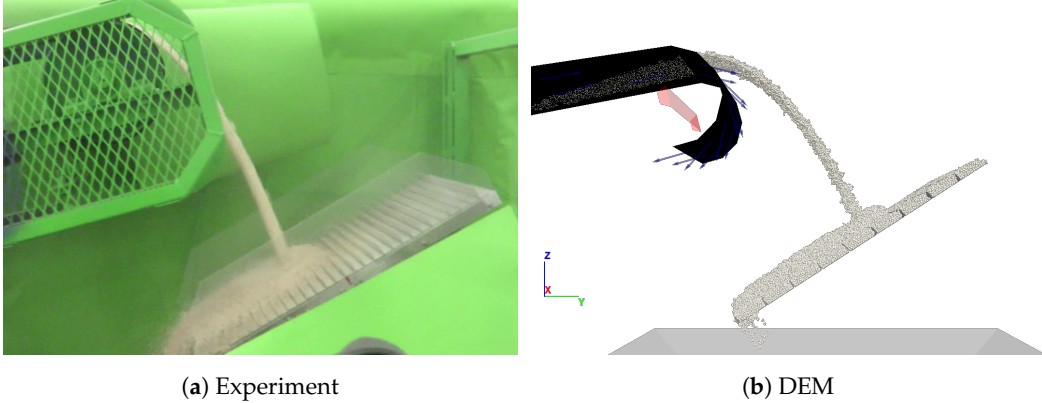

(**a**) Experiment      (**b**) DEM

**Figure 3.** Material flow from the conveyor head pulley onto the impact plate.

To perform a single test, 20 kg of sand was evenly loaded onto the feeding conveyor, as shown in Figure 4a. The loading zone started 2.07 m behind the centre of the head pulley and stretched over a length of 2.09 m. This ensured that the material travelled at the same speed as the belt before reaching the point of discharge. The width of the loading zone was approximately 300 mm. The conveyor was mounted on four HBM U2A 10 kN load cells, sampled at 200 Hz, from which the mass flow rate could be calculated. The conveyor speed was always set to $1.35 \, \text{m s}^{-1}$.

Three measures of the sand's bulk behaviour were identified to validate the model. Firstly, the maximum (peak) force exerted on the impact plate during discharge, the material mass that remained (residual weight) on the impact plate after the material stopped flowing, and lastly, the shape and height of the build-up pile after discharge.

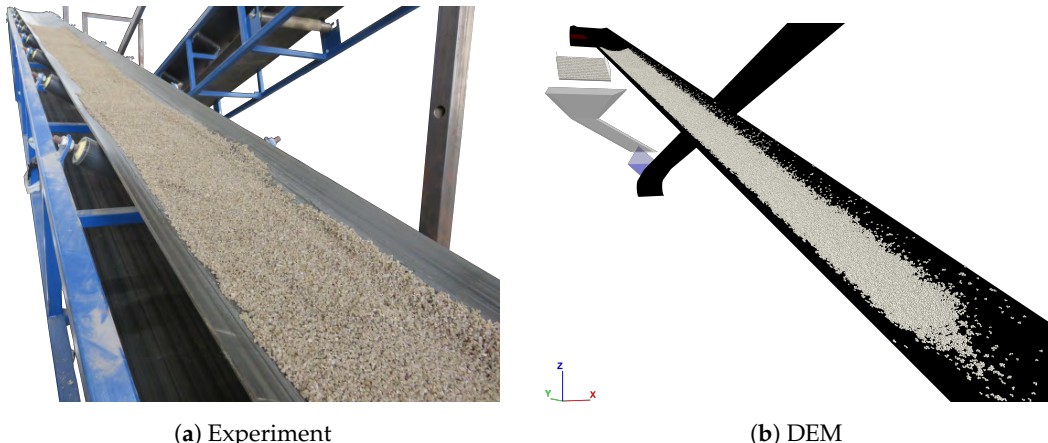

(**a**) Experiment                    (**b**) DEM

**Figure 4.** Loading of test material on the conveyor.

### 2.3. DEM Model

The model of the geometry of the conveyor, including the belt and transition zone, the head pulley and the impact plate, was an accurate representation of the experiment. It was assumed that the flexibility of the belt was negligible, and therefore, it was modelled as rigid. For a discussion on applications where it is important to include the flexibility of the belt, see [43]. The feeding conveyor was loaded with material in identically the same way as in the experiment (Figures 3b and 4b), and with the particle density accurately calibrated, it could be assumed that the initial bulk density of the material in the model was the same as in the experiment.

The particles were modelled using multi-sphere particles called clumps (following the terminology used in PFC [46], the software used in this study). Ten sand grains were randomly selected from the coarse sample and scanned to produce a detailed stereolithography (STL) model of each (the medium and fine grades having particles too small to scan). To produce the clumps, a bubble pack algorithm was used to optimally fill each STL model with three constituent spheres. As an example, three of the ten particles are shown in Figure 5.

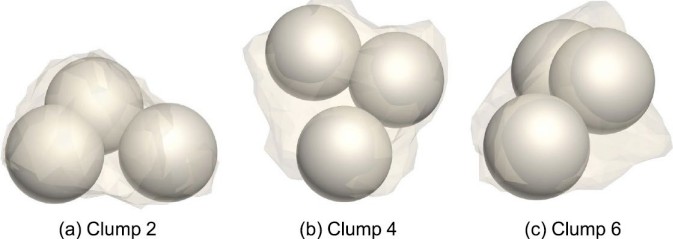

(a) Clump 2          (b) Clump 4          (c) Clump 6

**Figure 5.** Three of the ten clumps generated based on the coarse-grade sand. Reproduced with permission from [44]; published by Elsevier, 2023

It was assumed that the other sand grades had similar shapes, and therefore, these ten clumps were used to model all three grades. As a result, the unscaled clumps modelled the coarse-grade sand at a scale of 1.0, and it modelled the medium grade at a scale of 2.5 and the fine grade at a scale of 16.5. A clump naming convention (key) was introduced, as shown in Table 1, where the unscaled clumps were represented by C1.0M2.5F16.5, for example, based on the scale at which it modelled the individual grades: 'C' for coarse, 'M' for medium and 'F' for fine. To investigate the effects of particle scaling, the generated set of clumps was scaled by a factor of 1.3, 1.5, 2.0, 3.0 and 4.0, respectively, for which the corresponding scales for the medium and fine grades are listed in Table 1. For each scale, the mean particle diameter $\bar{d}_p$ is also shown.

**Table 1.** Clump naming convention (key), mean diameter and scaling factor for each of the three grades.

| Clump Key | Mean Diameter $\bar{d}_p$ [mm] | Sand Grade Scale Factor | | |
|---|---|---|---|---|
| | | Coarse | Medium | Fine |
| C1.0M2.5F16.5 | 4.75 | 1.0 | 2.5 | 16.5 |
| C1.3M3.2F21.4 | 6.18 | 1.3 | 3.2 | 21.4 |
| C1.5M3.7F24.7 | 7.13 | 1.5 | 3.7 | 24.7 |
| C2.0M5.0F33.0 | 9.50 | 2.0 | 5.0 | 33.0 |
| C3.0M7.4F49.5 | 14.25 | 3.0 | 7.4 | 49.5 |
| C4.0M9.9F60.0 | 19.00 | 4.0 | 9.1 | 60.0 |

For a detailed calibration of each of the six sets of scaled clumps for each of the three sand grades, see Scheffler and Coetzee [45]. The DEM software package, PFC [46], and the built-in linear cohesion contact model were used, and for clarity, the input parameter values used in this study are summarised in Appendix A. Liquid migration might be present in the experiment; however, it was not included in the model, but should be considered in future studies [47–49].

## 3. Results and Discussion

### 3.1. Experimental Results

Three bulk measures, as a function of the percentage saturation (moisture content), are shown in Figure 6. These include the residual weight of material remaining on the impact plate after discharge, the peak impact force during discharge and the residual pile height taken as the maximum height measured perpendicular to the plate's face.

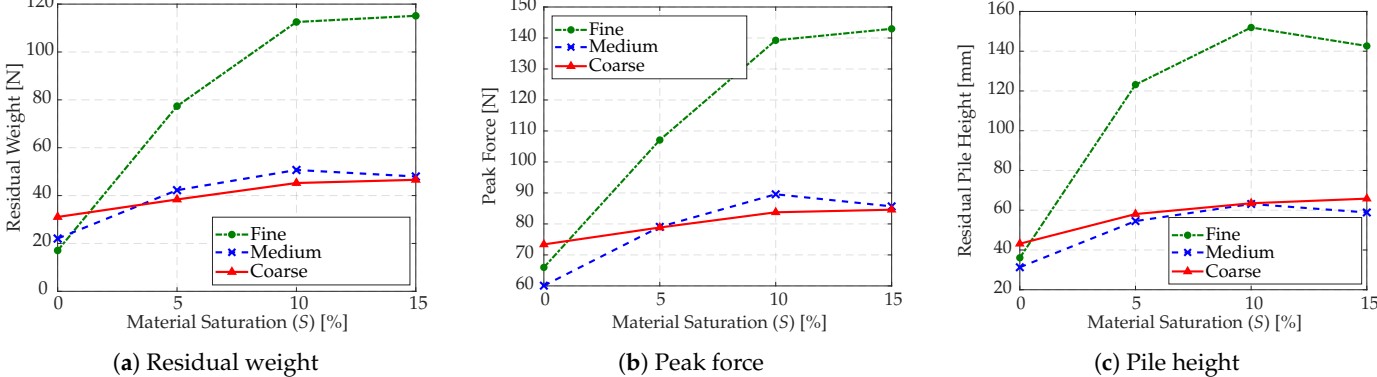

(**a**) Residual weight  (**b**) Peak force  (**c**) Pile height

**Figure 6.** Experimental results of the three sand grades with varying saturation levels, *S*.

Figure 6a shows how the residual weight, in a dry state ($S = 0\%$), increased as the particle size increased from fine to coarse. This corresponded with the same trend observed in the calibrated coefficient of friction, which was $\mu^{pp} = 0.20$, 0.30 and 0.35, respectively. Contrastingly, this trend was reversed when the wet materials were examined. The fine sand also showed the largest increase in residual weight with increased moisture, whilst the medium sand's residual weight roughly doubled and the coarse sand's slightly increased. A similar response is observed in Figure 6b for the peak force. As expected, the trend in pile height, shown in Figure 6c, closely followed that of the residual weight.

In summary, the fine sand was significantly more sensitive to changes in the moisture content. The addition of liquid bridges and the resultant capillary forces lead to higher bond numbers (ratio of cohesion force to particle weight) when the particles are smaller. This was noticeable for saturation levels up to $S = 10\%$, after which there was no significant increase or even a slight decrease in the case of the pile height.

### 3.2. Qualitative Comparisons

A qualitative analysis of the build-up formation during a simulation is an excellent means of identifying obvious errors in the modelling process [50]. Wet-and-sticky materials tend to stick to walls and build up, which can ultimately result in blockages [40]. Thus, when evaluating the accuracy of a DEM model, one aspect to consider is the profile and size of the build-up (see [31,34,40,51] for example).

Figure 7 shows the experimental results for fine sand at four saturation levels, and Figure 8 shows the corresponding DEM results. The side profiles closely resembled those observed by Carr [40], Carr et al. [42], who referred to this as the 'rhino horn' due to its distinctive shape.

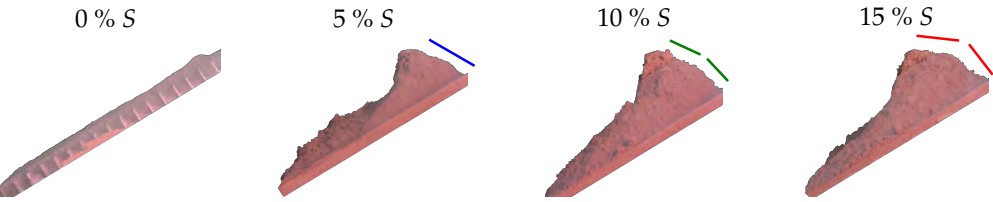

**Figure 7.** Experimental pile of the fine sand at four saturation states, *S*.

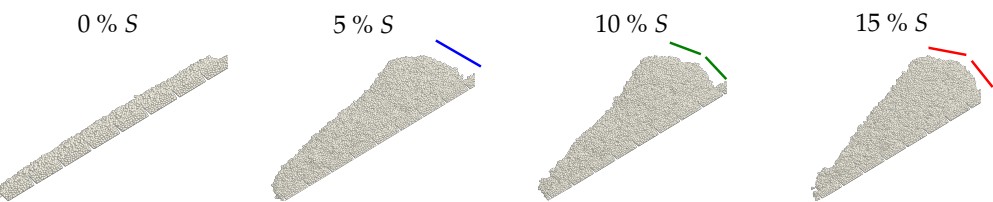

**Figure 8.** Modelled pile of fine sand at four saturation states, modelled with a clump scale of F21.4.

Interestingly, as the degree of saturation was increased from 5% to 15%, the profile became progressively more perpendicular to the impact plate at the end furthest away from the head pulley. That is, a kink developed with two distinct inclines, as indicated by the added lines in Figure 7. The model accurately predicted this phenomenon, as indicated in Figure 8. However, it failed to accurately predict the profile in the region where the material flowed from the plate (towards the left hand side of the plate in the figures). This is further addressed in Section 3.4.

### 3.3. Quantitative Comparisons

The impact force–time response was compared for the three sand grades, of which an example is shown in Figure 9 for the medium sand and six clump scale factors at four saturation levels. The total loading force resulted from a combination of net material build-up on the plate and the dynamic impact force due to the discharging stream. Thus, the peak loading force was reached just before the discharge stopped (after approximately 2.5 s to 2.8 s). Hereafter, the net material build-up decreased as some material continued to drain from the plate. After material drainage stopped, a residual amount of material was retained on the plate (the residual weight).

In DEM, relatively good results were achieved for the three smallest clump scale factors (M2.5 to M3.7) for all the saturation states. However, the load-time response was, in general, more gradual and peaked slightly later than the experimental response. While Figure 9 shows the results of the medium sand, the same analysis was performed for the coarse and fine sands. With the fine sand, scaling factors ranging from F16.5 to S49.5 produced accurate results. This indicates that higher levels of cohesion (that of the fine sand versus the medium sand for identical saturation levels) allows for a larger degree of particle upscaling.

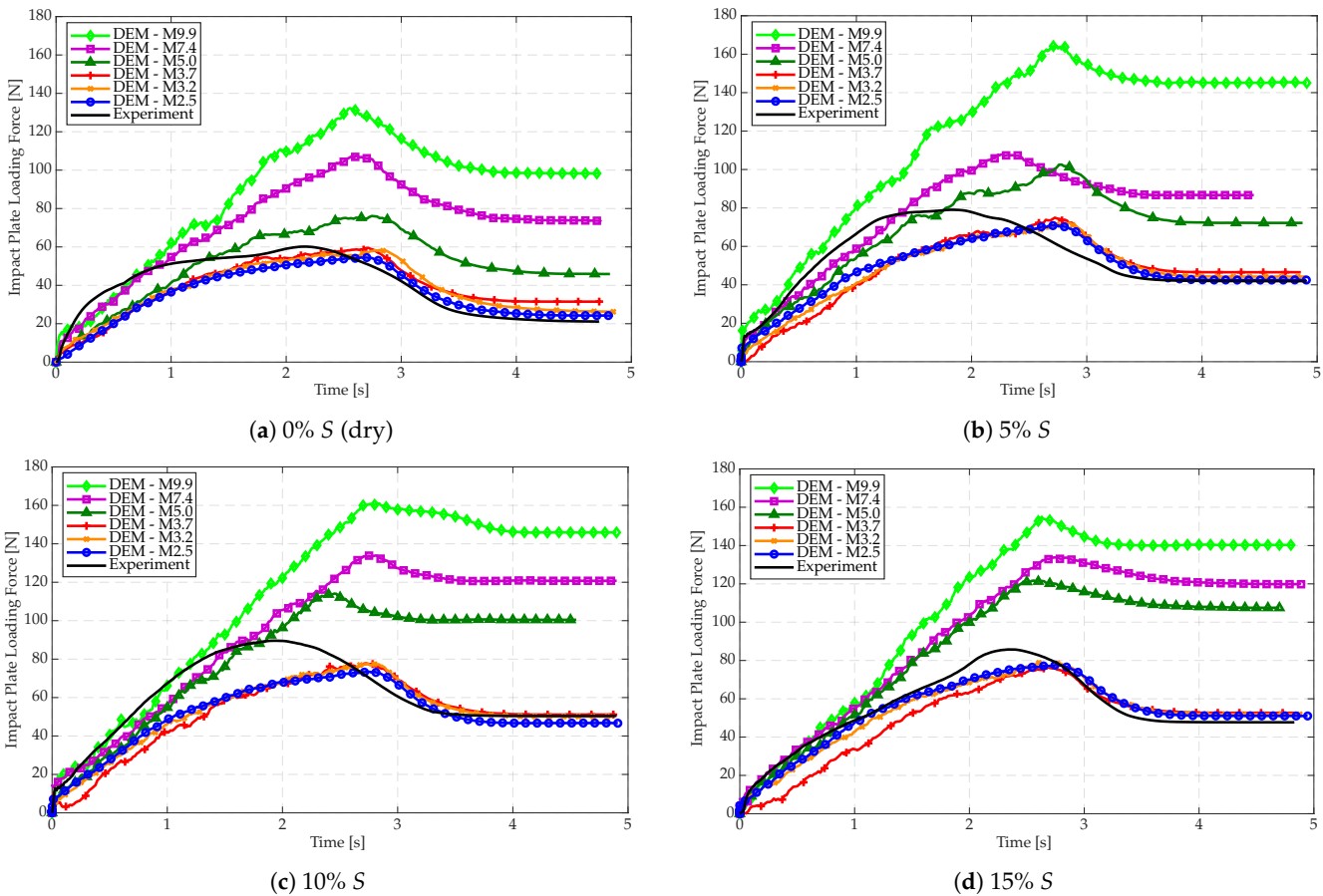

**Figure 9.** Comparison of the impact force–time response of the medium sand, showing the experimental results and the DEM results for six clump scale factors for each of the four saturation levels.

The remainder of this section presents a more detailed investigation of the peak impact force and the residual weight taken from the results in Figure 9. The experimental mean values are reported, and modelled results were considered accurate if within ±15% of the experimental mean value. As reference, Gröger and Katterfeld [28] considered wall impact forces to be accurate if the error was <17%, while Rossow and Coetzee [34] considered an error margin ≤10% to be accurate. Xie et al. [52] considered errors < 20% to be acceptable when modelling wear processes in a conveyor transfer system.

### 3.3.1. Comparison of the Peak Force

Figure 10 shows the modelled peak force of the coarse sand as a function of the clump scale for four saturation levels. The peak force increased with an increase in the clump scale, and accurate predictions were made for clump scales up to C1.5, i.e., a mean particle diameter $\bar{d}_p = 7.13$ mm (Table 1). Using C2.0 slightly over-predicted the peak force for the two higher saturation levels (Figure 10c,d). This result is in agreement with the observations made by Rossow and Coetzee [34], who reported a maximum scaling factor of 1.6 when modelling the flow of non-cohesive material through a transfer chute, whilst clump scale factors as large as 2.0 may still be acceptable for indicative purposes in certain applications.

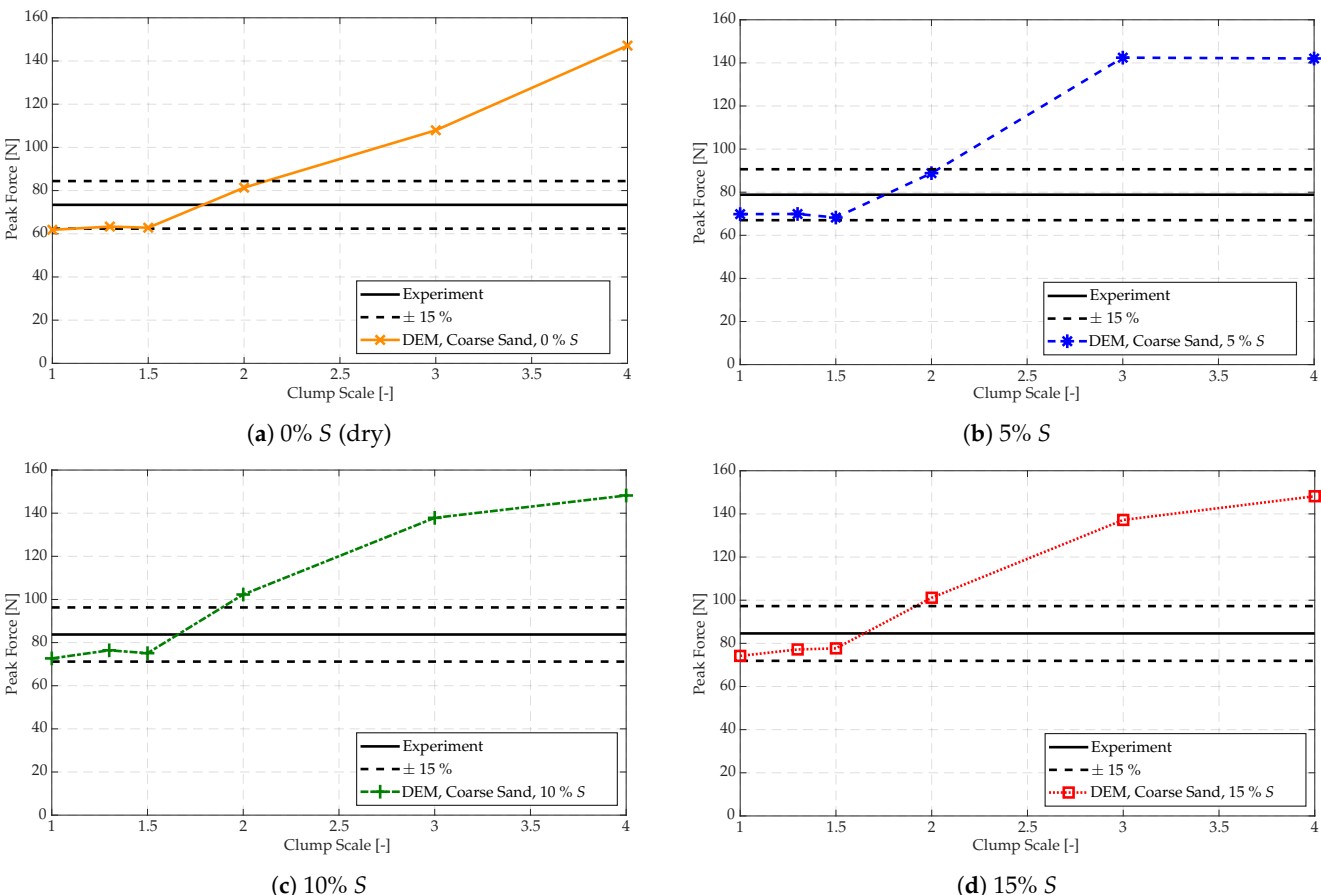

**Figure 10.** DEM prediction of the peak force of the coarse sand as a function of the clump scale and four saturation levels. The experimental mean ($\pm$15%) is indicated for each saturation level.

Similar trends were observed for the medium sand; however, clump scaling factors up to M3.7 were accurate, i.e., a mean particle diameter $\bar{d}_p = 7.13$ mm (Table 1). The peak forces of the fine sand are shown in Figure 11. In the dry case (Figure 11a), accurate predictions were made for clump scales up to a factor of F24.7, i.e., a mean particle diameter $\bar{d}_p = 7.13$ mm (Table 1). However, in the wet cases (Figure 11b–d), the predictions were accurate up to a scale factor of F49.5, i.e., a mean particle diameter $\bar{d}_p = 14.25$ mm (Table 1).

The discharging stream can be defined in terms of its thickness, as shown in Figure 12, and its width which is approximately equal to the belt width. Usually, the stream thickness is far less than its width (also mentioned by [24]). Therefore, it is postulated here that the stream thickness dictates the modelled resolution required to accurately predict the forces exerted on the impact plate.

In the experiments, the stream's thickness was approximately 30 mm for both the coarse and medium sand grades and not very sensitive to changes in moisture content. The maximum clump scale factors for these two sand grades were C1.5 and M3.7, respectively, for all saturation levels, Figure 10. Both of these scale factors equate to a mean particle diameter of $\bar{d}_p = 7.13$ mm (Table 1). Thus, a resolution of at least $^{30}/_{7.13} \approx 4$ particle diameters was needed across the thickness of the discharge stream to accurately model the impact force.

This is further supported by the results of the fine sand where, in the dry state, the thickness of the discharge stream was also approximately 30 mm, but in the wet states, it was approximately 60 mm (Figure 12). In the dry state, the maximum scale factor was F24.7 (Figure 11a), which equates to a mean particle diameter of $\bar{d}_p = 7.13$ mm, and again a stream thickness resolution of $^{30}/_{7.13} \approx 4$ particle diameters. In the wet states, the maximum scale factor was F49.5 (Figure 11b–d), which equates to a mean

particle diameter of $\bar{d}_p = 14.25\,\text{mm}$, and again a stream thickness resolution of $^{60}/_{14.25} \approx 4$ particle diameters.

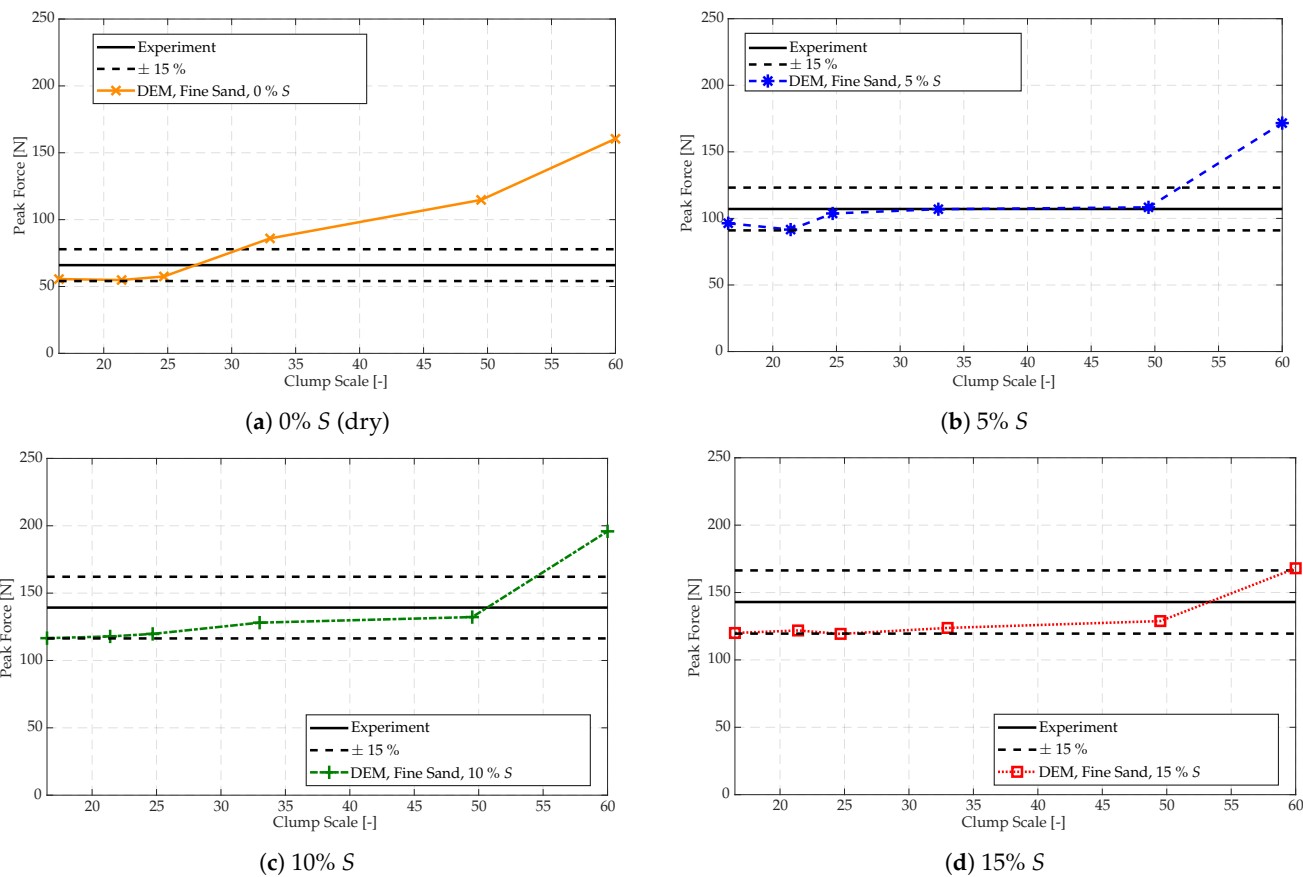

**Figure 11.** DEM prediction of the peak force of the fine sand as a function of the clump scale and four saturation levels. The experimental mean ($\pm15\%$) is indicated for each saturation level.

Further analysis of the results presented by Rossow and Coetzee [34] showed that the maximum scaling factor, for the accurate modelling of corn grain flow through a transfer chute, resulted in a stream thickness resolution of at least four particle diameters. Furthermore, Grima and Wypych [53] modelled the impact forces of polyethene pellets and, further analysing their results, showed that a stream thickness resolution of at least $2.8 \approx 3$ particle diameters (corresponding to a particle scale factor of 2) was needed.

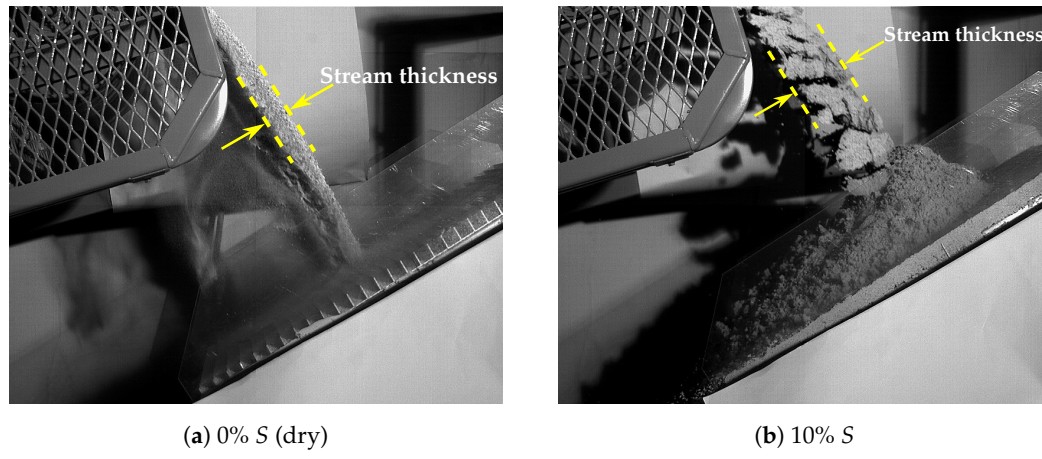

**Figure 12.** Comparison of the fine sand's dry and wet discharging stream thickness.

### 3.3.2. Comparison of the Residual Weight

The residual weight of the coarse grade sand is shown in Figure 13 as a function of the clump scale for the four saturation levels. The general trend was similar to that of the peak impact force (Figure 10), and again the maximum accurate scale factor was approximately C1.5 to C2.0.

The medium sand showed trends similar to that of the coarse sand, and the results of the fine sand are given in Figure 14. For the dry state, all the scale factors over-predicted the residual weight; however, similar to the peak force, scale factors up to F24.7 provided adequate results. For a saturation $S = 5\%$, the maximum scale factor was also approximately F24.7, while for $S = 10\%$ and $S = 15\%$, it was F49.5. Compared to the peak force results (Figure 11, the most noticeable difference in their trends is that for $S = 5\%$, the maximum scale factor is reduced. This indicates that, for the same material, but different flow mechanisms or applications, different maximum scale factors may apply for the same material.

### 3.4. Pile Formation Sensitivity

The qualitative assessment in Section 3.2 showed that the model could, in general, accurately predict the peak (maximum) height and general shape of the material pile. However, comparing Figure 8 with Figure 7, it can be seen that an exact prediction of the pile's profile of the cohesive material could not be achieved, with the peak (apex) of the pile being less acute in the model than in the experiment. A clump scale factor of F21.4 was used in Figure 8, and increasing the scale factor to F24.7 produced even more rounded peaks and flatter material drainage, as shown in Figure 15. A similar observation was made by Carr [40], who used the residual weight on an impact plate to calibrate the particle-wall cohesivity parameters. That is, the residual weight could be successfully calibrated, but the shape or profile of the pile (build-up) could be significantly different. Thus, the sensitivity of the pile's profile to the contact model and the PSD was further investigated. The fine sand was used, and the clump scale factor was set to F24.7.

While a linear cohesion contact model was used to produce all of the results presented thus far, it was repeated using the Johnson–Kendall–Roberts (JKR) model [54] and a simplified JKR (SJKR) model. The JKR model is an extension of the well-known Hertz contact model which incorporates a cohesion force which is based on van der Waals effects and acts between two particles while in contact, as well as over a short separation gap. Moisture-induced cohesion, as studied here, is due to liquid bridges and capillary forces and not van der Waals effects. However, the JKR model has been used in a number of studies to model the cohesive behaviour of moist granular materials [38,55–57]. The specific implementation of the JKR model into the PFC software that was used in this study can be found here [58]. Due to implementation issues and computation costs, simplified versions of the JKR model were developed. These models include a cohesion force, but make use of a simplified contact area and exclude any short-ranged forces (particle have to make physical contact for the force to be active). In this study, the SJKR-A model was used as described and implemented by Coetzee [59].

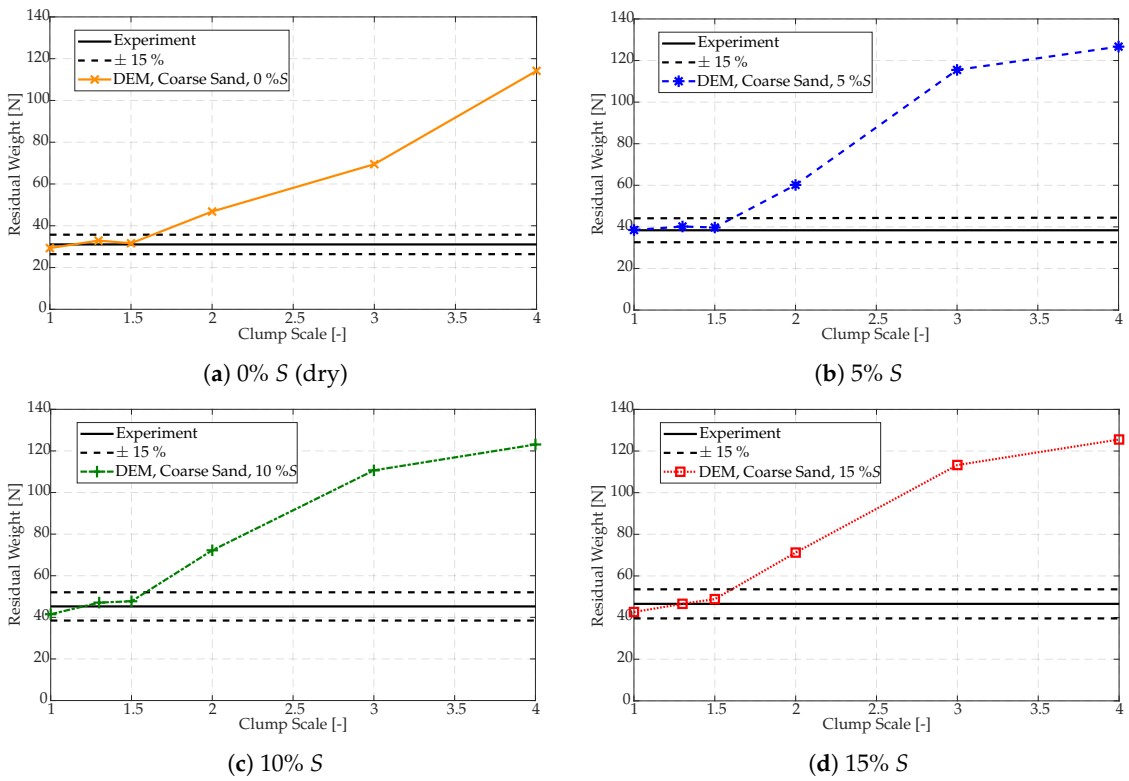

**Figure 13.** DEM prediction of the residual weight of the coarse sand as a function of the clump scale and four saturation levels. The experimental mean (±15%) is indicated for each saturation level.

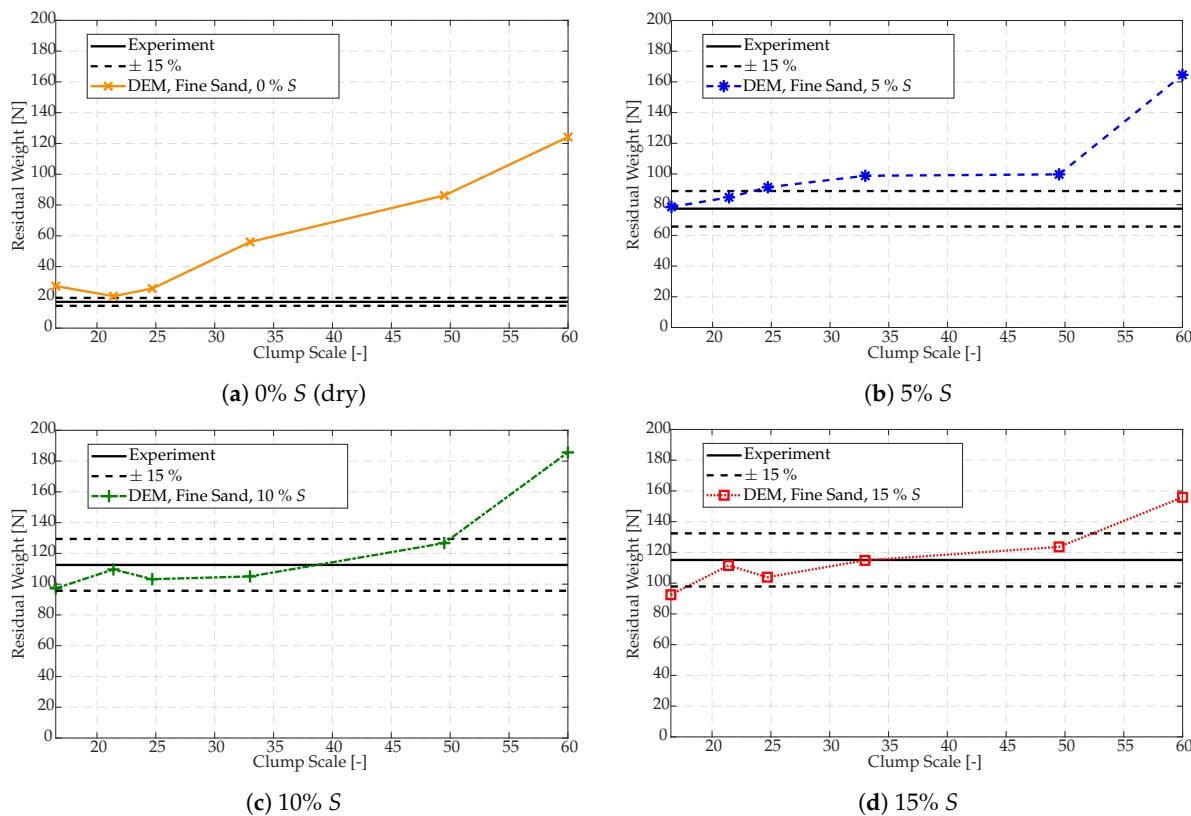

**Figure 14.** DEM prediction of the residual weight of the fine sand as a function of the clump scale and four saturation levels. The experimental mean (±15%) is indicated for each saturation level.

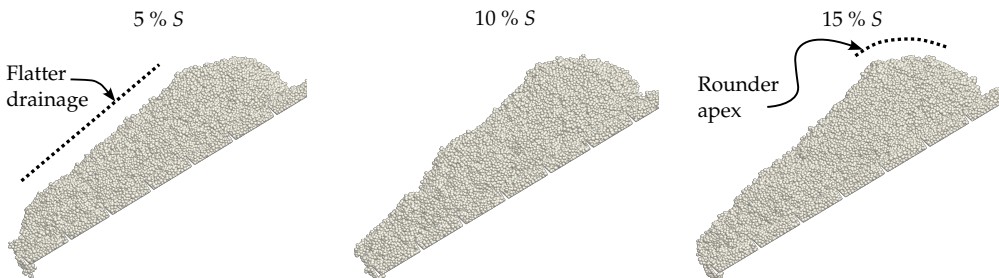

**Figure 15.** Modelled pile of fine sand at four saturation states, modelled with a clump scale of F24.7.

To compare the results from these models to that of the linear cohesion model, the applicable parameter values were set to be equivalent to that of the linear cohesion model (Table A4 for the F24.7 clumps) and to have a similar pull-off (maximum cohesion) force. The resulting piles are shown in Figure 16. These models, however, did not improve the results, and the resulting piles were still too small. Similar results were obtained by Carr et al. [42], who tested a liquid bridge model and an SJKR model. It was concluded that the SJKR model could not accurately model particle–particle cohesion and resulted in build-up that was too small. On the other hand, the liquid bridge model was significantly more accurate. One reason for this is that the SJKR model does not allow for short-ranged forces, and contacts break the moment the two particles loses physical contact. In the liquid bridge model, however, the volume of the liquid bridge determines the pull-off distance, which, when larger than zero, allows for short-ranged forces. The linear cohesion model used in this study is basically a linearised liquid bridge model of which the pull-off distance is an independent input parameter, and that might be the reason why it outperforms the SJKR model.

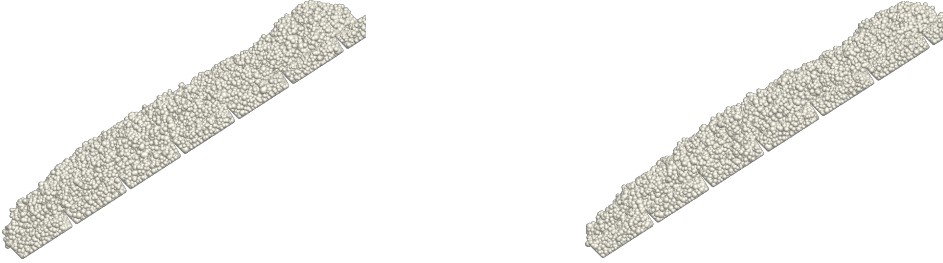

**Figure 16.** Modelled pile of fine sand (F24.7) using the JKR (**left**) and SJKR-A (**right**) models.

Lastly, the effect of the PSD on the formation of the build-up pile was investigated. Due to practical reasons and for simplicity, a selection of coarse sand grains was scanned and used for all the grades, in other words, also for the medium and fine grades. This also meant that the coarse grade's PSD, shown in Figure 1a, was used for the medium and fine grades (and scaled accordingly based on the mean particle diameter). In an attempt to improve the results, the scanned particles from the coarse grade were used (the grains from the fine grade were too small to scan), but re-sampled using the fine grade's PSD (Figure 1c). This newly generated PSD was then scaled to ensure that the mean particle diameter equalled that of the previous PSD scaled by a factor F24.7.

All the other input parameters were kept constant, and the resulting piles are shown in Figure 17. Comparing this to Figures 7 and 15 shows the improvement in modelling the profile of the build-up. Using the newly generated PSD, the triangular shape of the pile improved, and a sharper peak (apex) was predicted, which compared better to the experiment than the results from using the original PSD. Moreover, the material drainage towards the toe of the pile appeared less flat, and a kink manifested in the drainage profiles, which was similar to that in the experiment and visible in Figure 7. Consequently, it shows that using a more precise PSD with more accurate relative ratios between particle sizes was

vital for accurate simulations, even if the whole PSD was significantly scaled relative to the physical material.

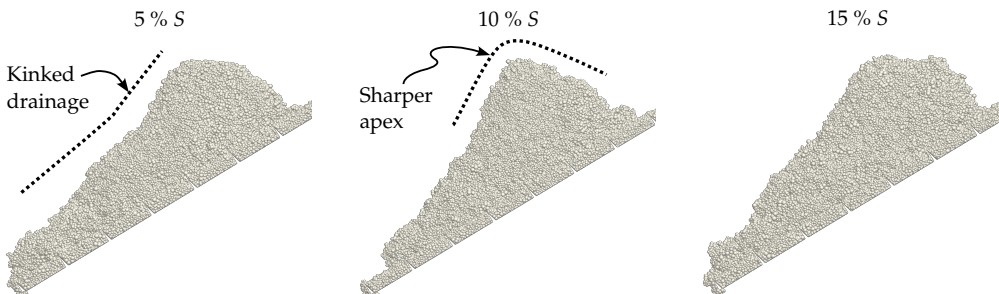

**Figure 17.** Modelled pile of the fine sand for three saturation states and a clump scale of F24.7 with the newly generated PSD based on fine sand.

### 3.5. Computation Time

The computation time is proportional to the size of the time step and the number of particles (and contacts) in the model. The time step size is proportional to $\sqrt{m/k}$, where $m$ is the effective contact mass (based on the mass of the two particles in contact) and $k$ is the contact stiffness. The contact stiffness was kept constant, and the particle mass increased with the scaling factor cubed, $m \propto \alpha^3$. Automatic time stepping was used, i.e., the software selected the time step size based on the contact mass and stiffness and automatically adjusted it when the particles were scaled.

Furthermore, scaling the particle size by a factor $\alpha$ decreased the number of particles in the model by a factor $\alpha^3$. Thus, with an increase in the particle size, the computation time decreased due to a significant reduction in the number of particles (and contacts) in the model, but also due to an increase in the time step. Time studies showed that the computation time decreased by a factor of approximately $\alpha^4$, similar to the findings by [60].

### 4. Conclusions

The discharge of moist cohesive sand from a conveyor onto an impact plate was analysed. Experiments were performed with three grades of sand, each with a different particle size distribution (mean particle diameters of 4.75 mm, 1.92 mm and 0.29 mm) and four levels of saturation (moisture content). The discrete element models used multi-sphere particles which were calibrated in a related study for each of the sand grades and saturation levels. Furthermore, particle upscaling (coarse graining) was used, and the modelled impact force and the material build-up on the plate compared to measurements, both qualitatively and quantitatively.

Qualitatively, the material build-up was modelled reasonably well. Quantitatively, the model accurately predicted the peak force during discharge and the residual weight on the plate at the end of discharge (error < 15%), provided that the particles were not excessively scaled. The maximum scale factor was determined by the modelled resolution of the discharge stream, and was independent of the sand grade and moisture level. With at least four particle diameters across the stream thickness, the model was accurate. It is concluded that the particle scaling factor is limited by the characteristic length of the bulk flow mechanism (the thickness of the discharge stream in this case) and not the size of the physical particles. For a given grade of sand, the characteristic lengths increased with an increase in cohesion (moisture), and so did the maximum scale factor.

These results are valuable in the modelling of cohesive materials in large-scale industrial applications, where wet-and-sticky materials can cause build-up and blockages. However, the accuracy of the model should in future be validated for other cohesive materials. Furthermore, upscaled particles can significantly reduce computational cost. With the particle size scaled by a factor $\alpha$, the number of particles in the model decreased by a factor $\alpha^3$ and the computation time by a factor of approximately $\alpha^4$.

**Author Contributions:** Conceptualisation, C.J.C. and O.C.S.; methodology, C.J.C. and O.C.S.; validation, O.C.S.; writing—original draft preparation, O.C.S.; writing—review and editing, C.J.C.; supervision, C.J.C.; funding acquisition, C.J.C. All authors have read and agreed to the published version of the manuscript.

**Funding:** This research was funded by the National Research Foundation (NRF), grant number 137952.

**Data Availability Statement:** Data are contained within the article.

**Conflicts of Interest:** The authors declare no conflicts of interest. The funders had no role in the design of the study; in the collection, analyses, or interpretation of data; in the writing of the manuscript; or in the decision to publish the results.

## Abbreviations

The following abbreviations are used in this manuscript:

| | |
|---|---|
| DEM | Discrete Element Method, Discrete Element Model |
| JKR | Johnson–Kendall–Roberts |
| PFC | Particle Flow Code |
| PIV | Particle Image Velocimetry |
| PSD | Particle Size Distribution |
| SJKR | Simplified Johnson–Kendall–Roberts |
| SJKR-A | Simplified Johnson–Kendall–Roberts model-A |
| WSM | Wet-and-Sticky Material |

## Appendix A

The bulk calibration of the input parameters of the linear cohesive contact model, for the three sand grades, is documented in Scheffler and Coetzee [45]. The contact stiffness and damping were identical for the three sand grades and are given in Table A1. Here, the stiffness is given by $k$ and the critical damping ratio by $\beta$, with the subscripts 'n' and 's' indicating the contact normal and shear (tangential) directions, respectively, and the superscripts 'pp' and 'pw' indicating particle–particle and particle–wall contacts, respectively.

The other input parameters are given in Tables A2–A4 for the three sand grades, respectively. The particle density is designated as $\rho_p$, $\mu$ is the coefficients of friction, $F_0$ is the maximum pull-off (cohesion) force and $D_0$ the rupture or pull-off distance (separation gap where the contact breaks), and $S$ is the percentage of saturation.

**Table A1.** The contact stiffness and damping for all three sand grades.

| Parameter | Clump Scales | | | | | |
|---|---|---|---|---|---|---|
| | C1.0M2.5F16.5 | C1.3M3.2F21.4 | C1.5M3.7F24.7 | C2.0M5.0F33.0 | C3.0M7.4F49.5 | C4.0M9.9F60.0 |
| | Contact stiffness, $[\mathrm{kN\,m^{-1}}]$ | | | | | |
| $k_n^{pp}$ | 66 | 77 | 82 | 160 | 260 | 430 |
| $k_s^{pp}$ | 44 | 51 | 55 | 110 | 170 | 290 |
| $k_n^{pw}$ | 130 | 150 | 160 | 320 | 520 | 860 |
| $k_s^{pw}$ | 86 | 100 | 100 | 210 | 340 | 570 |
| | Contact damping, $[-]$ | | | | | |
| $\beta_n^{pw}$ | 0.30 | 0.30 | 0.30 | 0.30 | 0.30 | 0.30 |
| $\beta_n^{pp}$ | 0.36 | 0.36 | 0.36 | 0.36 | 0.36 | 0.36 |

**Table A2.** Coarse sand input parameters.

| Parameter | | Clump Scales | | | | | |
|---|---|---|---|---|---|---|---|
| | | **C1.0** | **C1.3** | **C1.5** | **C2.0** | **C3.0** | **C4.0** |
| | | Particle density, $[\mathrm{kg\,m^{-3}}]$ | | | | | |
| $\rho_\mathrm{p}$ | | 2480.0 | 2500.0 | 2540.0 | 2530.0 | 2575.0 | 2650.0 |
| | | Coefficients of friction, $[-]$ | | | | | |
| $\mu^\mathrm{pw}$ | | 0.36 | 0.36 | 0.36 | 0.36 | 0.36 | 0.36 |
| $\mu^\mathrm{pp}$ | | 0.35 | 0.35 | 0.35 | 0.35 | 0.35 | 0.35 |
| | | Cohesivity parameters *, $F_0$ [mN], $D_0$ [mm] | | | | | |
| 0% $S$ | $F_0^\mathrm{pp}$ | 0.12 | 0.45 | 0.45 | 0.56 | 1.6 | 4.0 |
| | $D_0^\mathrm{pp}$ | 0.0 | 0.0 | 0.0 | 0.0 | 0.0 | 0.0 |
| 5% $S$ | $F_0^\mathrm{pp}$ | 0.24 | 0.86 | 0.80 | 3.75 | 12.8 | 30.5 |
| | $D_0^\mathrm{pp}$ | 1.0 | 0.6 | 0.8 | 0.39 | 0.7 | 0.55 |
| 10% $S$ | $F_0^\mathrm{pp}$ | 0.24 | 0.70 | 0.80 | 3.75 | 12.8 | 30.0 |
| | $D_0^\mathrm{pp}$ | 1.3 | 0.7 | 0.9 | 0.4 | 0.8 | 0.64 |
| 15% $S$ | $F_0^\mathrm{pp}$ | 0.24 | 0.70 | 0.80 | 3.75 | 12.8 | 30.0 |
| | $D_0^\mathrm{pp}$ | 1.5 | 0.8 | 1.0 | 0.45 | 0.9 | 0.7 |

* with $F_0^\mathrm{pw} = 2 \times F_0^\mathrm{pp}$ and $D_0^\mathrm{pw} = 0.8 \times D_0^\mathrm{pp}$

**Table A3.** Medium sand input parameters.

| Parameter | | Clump Scales | | | | | |
|---|---|---|---|---|---|---|---|
| | | **M2.5** | **M3.2** | **M3.7** | **M5.0** | **M7.4** | **M9.9** |
| | | Particle density, $[\mathrm{kg\,m^{-3}}]$ | | | | | |
| $\rho_\mathrm{p}$ | | 2600.0 | 2614.5 | 2625.0 | 2650.0 | 2674.5 | 2690.0 |
| | | Coefficients of friction, $[-]$ | | | | | |
| $\mu^\mathrm{pw}$ | | 0.36 | 0.36 | 0.36 | 0.36 | 0.36 | 0.36 |
| $\mu^\mathrm{pp}$ | | 0.30 | 0.30 | 0.30 | 0.30 | 0.30 | 0.30 |
| | | Cohesivity parameters *, $F_0$ [mN], $D_0$ [mm] | | | | | |
| 0% $S$ | $F_0^\mathrm{pp}$ | 0.25 | 0.1 | 1.2 | 1.5 | 6.0 | 19.0 |
| | $D_0^\mathrm{pp}$ | 0.0 | 0.0 | 0.0 | 0.0 | 0.0 | 0.0 |
| 5% $S$ | $F_0^\mathrm{pp}$ | 0.30 | 1.0 | 1.4 | 4.2 | 21.0 | 67.5 |
| | $D_0^\mathrm{pp}$ | 0.9 | 0.5 | 0.8 | 0.5 | 0.3 | 0.6 |
| 10% $S$ | $F_0^\mathrm{pp}$ | 0.35 | 1.0 | 1.7 | 4.2 | 21.0 | 67.5 |
| | $D_0^\mathrm{pp}$ | 1.0 | 0.8 | 0.9 | 0.9 | 0.8 | 1.0 |
| 15% $S$ | $F_0^\mathrm{pp}$ | 0.38 | 1.0 | 2.1 | 4.7 | 21.0 | 50.0 |
| | $D_0^\mathrm{pp}$ | 1.4 | 1.0 | 1.0 | 1.0 | 1.3 | 1.0 |

* with $F_0^\mathrm{pw} = 2 \times F_0^\mathrm{pp}$ and $D_0^\mathrm{pw} = 0.8 \times D_0^\mathrm{pp}$

**Table A4.** Fine sand input parameters.

| Parameter | | Clump Scales | | | | | |
|---|---|---|---|---|---|---|---|
| | | F16.5 | F21.4 | F24.7 | F33.0 | F49.5 | F60.0 |
| | | Particle density, $[\mathrm{kg\,m^{-3}}]$ | | | | | |
| $\rho_{\mathrm{p}}$ | | 2870.0 | 2900.0 | 2934.00 | 2950.0 | 3000.0 | 3020.0 |
| | | Coefficients of friction, $[-]$ | | | | | |
| $\mu^{\mathrm{pw}}$ | | 0.36 | 0.36 | 0.36 | 0.36 | 0.36 | 0.36 |
| $\mu^{\mathrm{pp}}$ | | 0.20 | 0.20 | 0.20 | 0.20 | 0.20 | 0.20 |
| | | Cohesivity parameters *, $F_0$ [mN], $D_0$ [mm] | | | | | |
| 0% $S$ | $F_0^{\mathrm{pp}}$ | 1.0 | 0.8 | 1.5 | 10.0 | 37.5 | 62.0 |
| | $D_0^{\mathrm{pp}}$ | 0.0 | 0.0 | 0.0 | 0.0 | 0.0 | 0.0 |
| 5% $S$ | $F_0^{\mathrm{pp}}$ | 1.40 | 4.0 | 9.5 | 19.0 | 100.0 | 195.0 |
| | $D_0^{\mathrm{pp}}$ | 1.9 | 1.9 | 0.7 | 1.0 | 0.9 | 1.0 |
| 10% $S$ | $F_0^{\mathrm{pp}}$ | 1.41 | 6.0 | 9.5 | 18.0 | 103.3 | 195.0 |
| | $D_0^{\mathrm{pp}}$ | 2.0 | 1.7 | 1.4 | 1.7 | 1.0 | 1.2 |
| 15% $S$ | $F_0^{\mathrm{pp}}$ | 1.42 | 6.0 | 10.0 | 18.0 | 97.0 | 195.0 |
| | $D_0^{\mathrm{pp}}$ | 2.3 | 2.0 | 1.0 | 1.8 | 1.0 | 1.6 |

* with $F_0^{\mathrm{pw}} = 2 \times F_0^{\mathrm{pp}}$ and $D_0^{\mathrm{pw}} = 0.8 \times D_0^{\mathrm{pp}}$

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
