# Peer review of "Discrete Element Modelling of a Bulk Cohesive Material Discharging from a Conveyor Belt onto an Impact Plate"

_minerals, doi:10.3390/min13121501_

Round 1
Reviewer 1 Report
Comments and Suggestions for Authors
The authors present a comprehensive numerical study on cohesive granular materials discharge and their interaction with a conveyor belt and impact on a rigid plate, using DEM. The numerical results are compared with experimental data and errors less than 15% are considered acceptable by the authors. However, the impact-force time response looks qualitatively different, particularly at low moisture levels. One important contribution of this work is the appropriate resolutions, i.e., upscaling ratios for the coarse and fine sands, identified for varying saturation percentages. Lastly, the effect of particle size distribution on the pile formation is also investigated numerically. The manuscript is nicely written and enough details regarding the experiments and simulations are provided. However, I have a few concerns that the authors could address in the manuscript.
1. Sec. 2.3: The conveyor belt seems to be assumed as a rigid body in the DEm simulation. I am curious about how relevant the belt-particle interaction is to the granular flow dynamics and the impact force. The surface coupling between deformable structures and discrete particles can be easily achieved with FEM-DEM surface coupling [1] and flexible discrete elements [2]. I wonder why the author didn't consider belt-particle interactions in the model.
[1] H. Cheng, S. Luding, T. Weinhart. CG-enriched concurrent multi-scale modeling of dynamic surface interactions between discrete particles and solid continua. Acta Mechanica Sinica, 39 (1) (2023), Article 722218
[2] A. Effeindzourou, B. Chareyre, K. Thoeni, A. Giacomini, F. Kneib. Modelling of deformable structures in the general framework of the discrete element method. Geotext. Geomembranes, 44 (2) (2016), pp. 143-156
2. Sec. 2.4: The particles are modelled using multi-spheres, as shown in Fig.5. Their calibration is done using data from uniaxial compression and ring shear experiments, as reported in another paper. Can you comment on how these parameters apply to conditions like rapid, dense, and maybe collisional flows, during the transport on a conveyor belt?
3. Fig. 4b: How the particles are prepared in the simulations to achieve the same initial conditions (e.g., bulk density) as in the experiments?
4. The authors commented on liquid-bridge models in lines 172 and 291. I agree it is acceptable to use cohesive contact models to capture capillary forces. However, liquid bridge models typically include migration of the liquid volume as well. Please explain whether capillary liquid volume migration is taken into account in the DEM model presented.
5. Fig. 9: It seems the agreement between numerical and experiment results gets better as the saturation level increases. Any reason why this happens?
Reviewer 2 Report
Comments and Suggestions for Authors
The manuscript entitled "Discrete Element Modelling of a Bulk Cohesive Material Discharging from a Conveyor Belt onto an Impact Plate" explores the industrial application of discharging cohesive materials from a belt conveyor onto an inclined impact plate. Overall, the paper is well written detailing the application with clarity.
However, the more detailed description of the mechanics of the Discrete Element Modelling (DEM) used in the context of partial saturation, specifically focusing on cohesion due to capillary forces should be elaborated. A more insight into the mechanisms and a few important mathematical formulations related to utilized models should be added.
The authors could consider the paper titled "Discrete Element Model for the Failure Analysis of Partially Saturated Porous Media with Propagating Cracks Represented with Embedded Strong Discontinuities, CMAME, 2022." and refer to modelling aspects of partial saturation and cohesion in lattice particle models
The paper effectively describes the application, but improvements are needed in more precise description of the mechanics of the DEM model.
Reviewer 3 Report
Comments and Suggestions for Authors
This paper investigates in great detail the impact of particle size amplification on discrete element simulation results, improving simulation efficiency. However, the following opinions need to be considered:
1.The applicability of this method on other granular materials needs to be verifiedï¼›
2.The location for testing the impact force sensor should be givenï¼›
3.Please simplify the writing of the conclusion
4.What is the time and cost savings after particle scaling?
Round 2
Reviewer 2 Report
Comments and Suggestions for Authors
The authors did not take into account this reviewer's suggestions. Although the authors use commercial code, some basic formulations of the model are always required in the scientific paper.